# The Development of the Bengamides as New Antibiotics against Drug-Resistant Bacteria

**DOI:** 10.3390/md20060373

**Published:** 2022-05-31

**Authors:** Cristina Porras-Alcalá, Federico Moya-Utrera, Miguel García-Castro, Antonio Sánchez-Ruiz, Juan Manuel López-Romero, María Soledad Pino-González, Amelia Díaz-Morilla, Seiya Kitamura, Dennis W. Wolan, José Prados, Consolación Melguizo, Iván Cheng-Sánchez, Francisco Sarabia

**Affiliations:** 1Department of Organic Chemistry, Faculty of Sciences, University of Malaga, 29071 Málaga, Spain; cristinaalcala@uma.es (C.P.-A.); fedemu@uma.es (F.M.-U.); mgcastro@uma.es (M.G.-C.); jmromero@uma.es (J.M.L.-R.); pino@uma.es (M.S.P.-G.); amelia@uma.es (A.D.-M.); 2Faculty of Pharmacy, Campus de Albacete, Universidad de Castilla-La Mancha, 02008 Albacete, Spain; antonio.sanchezruiz@uclm.es; 3Department of Molecular Medicine, The Scripps Research Institute, San Diego, CA 92037, USA; skita@scripps.edu (S.K.); wolan@scripps.edu (D.W.W.); 4Department of Biochemistry, Albert Einstein College of Medicine, New York, NY 10641, USA; 5Center of Biomedical Research (CIBM), Institute of Biopathology and Regenerative Medicine (IBIMER), University of Granada, 18100 Granada, Spain; jcprados@ugr.es (J.P.); melguizo@ugr.es (C.M.); 6Department of Anatomy and Embriology, Faculty of Medicine, University of Granada, 18071 Granada, Spain; 7Instituto de Investigación Biosanitaria ibs, Granada, 18012 Granada, Spain; 8Department of Chemistry, University of Zurich, Winterthurerstrasse 190, 8057 Zurich, Switzerland

**Keywords:** bengamides, antibiotics, drug-resistant bacteria, antitumor agents, SAR

## Abstract

The bengamides comprise an interesting family of natural products isolated from sponges belonging to the prolific *Jaspidae* family. Their outstanding antitumor properties, coupled with their unique mechanism of action and unprecedented molecular structures, have prompted an intense research activity directed towards their total syntheses, analogue design, and biological evaluations for their development as new anticancer agents. Together with these biological studies in cancer research, in recent years, the bengamides have been identified as potential antibiotics by their impressive biological activities against various drug-resistant bacteria such as *Mycobacterium tuberculosis* and *Staphylococcus aureus*. This review reports on the new advances in the chemistry and biology of the bengamides during the last years, paying special attention to their development as promising new antibiotics. Thus, the evolution of the bengamides from their initial exploration as antitumor agents up to their current status as antibiotics is described in detail, highlighting the manifold value of these marine natural products as valid hits in medicinal chemistry.

## 1. Introduction

The bengamide family of natural products (**1**–**24**) constitutes a group of interesting and inspiring molecules that has elicited intense research directed toward their total synthesis, analogue design, and biological and medicinal studies due to their potent antitumor properties [1]. The isolation of the first members of bengamides, A (**1**) and B (**2**) was in 1986 by Crews/Quiñoá et al. [2] from an undescribed specimen of a sponge belonging to the *Jaspidae* family in Benga Lagoon (Fiji Islands). This finding was followed by the discoveries of other members (bengamides C-Q) from other sponges, including *Jaspis* cf. *coriacea* [3,4], *Jaspis digonoxea* [5], *Jaspis carteri* [6], *Jaspis splendens* [7], *Pachastrissa* sp. [8], *Dorypleres splendens* [9], *Stelleta* sp. [10] and *Calthropella* sp. [11]. Interestingly, the discoveries by Crews [12] and later by Brönstrup et al. [13] of the bengamides E (**5**), F (**6**), E’ (**7**), and F’ (**8**) from the terrestrial myxobacteria *Myxococcus virescens* strengthened the initial hypothesis that the bengamides were the result of a symbiotic interaction between the sponges and bacteria [14], a notion that was proposed in virtue of the presence of the end-chain isopropyl group, which is characteristic in bacterial fatty acids [15]. More recently, one new member of the bengamides, bengamide R (**19**), was isolated and characterized from *Jaspis splendens* sponges, collected in Mauritius, within a research programme directed towards the discovery and identification of new antimycobacterial agents from marine sources [16] (Table 1).

Structurally, the bengamides contain a unique molecular structure featured by a C-10 side chain possessing four contiguous hydroxyl groups and a terminal disubstituted *E*-olefin, linked to an aminocaprolactam moiety through an amide group, which is the main structural component that distinguishes the different members of these natural products [17]. In fact, according to the caprolactam ring, the bengamides are classified into two structural classes, which are the following: (a) Type I, which includes bengamides A-D, G–J, L–O, Y and Z, and that contains a hydroxylysine-derived caprolactam, bearing or not a lipidic chain; and (b) Type II that includes bengamides E, F, E’, F’ and P–S, that contains lysine-derived caprolactam. The cases of bengamide K (**23**) and isobengamide E (**24**) fall outside of this classification with the lack of the entire polyketide fragment possessing an *N*-formyl group instead of in **23**, and with a different link between the polyketide and the caprolactam fragments in the isomeric **24** (Figure 1).

Preliminary biological evaluations of the bengamides revealed their prominent antitumor properties, displaying cytotoxicities in the 1.0 nM–3.3 μM range for the IC_50_ values against human breast MDA-MB-435 carcinoma cells, producing the arrest of the cells at both the G_0_/G_1_ and G_2_/M interfaces of the cell cycle and, as a consequence, apoptosis of tumoral cells [18]. The mode of action was disclosed by means of proteomic studies [19], revealing that the bengamides inhibited methionine aminopeptidases types 1 and 2 (MetAP1 and MetAP2), which was further demonstrated by the isolation of the complex enzyme-bengamide and its subsequent X-ray analysis [19]. This analysis allowed for the determination of the mode of interaction of the bengamides at the active site of these important enzymes, with the following key interactions: (1) a critical dinuclear metal centre placed as a deep invagination in the surface of the enzyme is coordinated with the hydroxyl groups at C3, C4, and C5; (2) interaction of the terminal isopropyl group of the olefin at the hydrophobic pocket P1, which contains the residues Phe-219, His-382 and Ala-414; and (3) the allocation of the caprolactam ring at the hydrophilic pocket P2, located at the solvent-exposed surface (Figure 2). As a consequence of the inhibition of methionine aminopeptidases by the bengamides, the activity of the proto-oncogene *c*-Src, involved in the development, growth, progression, and metastasis of numerous human cancers, is remarkably decreased, producing a delay in cell-cycle progression [20]. Therefore, it was possible to establish a link between cancer and inhibition of the MetAPs through the proto-oncogene *c*-Src and likely other oncogenes, which represent substrates for both MetAP1 and MetAP2, and, in this way, to justify the antitumoral effect of the bengamides. Interestingly, the enzyme MetAP2 is the biological target of the very well-known antiangiogenic compounds fumagillin and ovalicin [21,22], and their selectivities against MetAP2 is a key factor that is being explored, not only for the treatment of cancer but also for the development of antiobesity agents [23]. Thus, the toxicity found in the bengamides is attributable to the lack of selectivity against both MetAPs [24,25], and a future direction in this research might be the design of MetAP2-selective bengamides [26].

Given the outstanding antitumor properties displayed by the bengamides, an intense research activity has been devoted in relation to the bengamides directed towards the analysis, characterization, and development of bengamide-based anticancer candidates with antiangiogenic properties through the design, synthesis, and biological evaluation of analogues. In fact, one of the most active analogues identified was the compound LAF-389 (see Figure 3) [27,28], which reached a phase I clinical trials, which began in 2000. However, finally, this clinical trial was discontinued, and the bengamide analogue was withdrawn for further clinical investigations due to unanticipated cardiotoxicity [29]. In particular, from a total of 33 patients treated with LAF-389, eight patients suffered severe cardiovascular toxicity, which was not predicted during the preclinical studies in rats and dogs. As mentioned above, the first biological evaluations of the bengamides confirmed that these compounds possessed a broad array of biological activities, including not only antitumor activities, but also antihelmintic and antibiotic activities. In fact, bengamides A (**1**) and B (**2**) showed to be active against *Streptococcus pyrogenes* with MIC (Minimum inhibitory concentration that prevents visible growth of the bacteria) values of 4 and 2 μg/mL. However, their antibiotic properties were not further explored. As we will discuss in the present review, the ability of the bengamides to inhibit methionine aminopeptidases was recently exploited for the case of the MetAP of the *Mycobacterium tuberculosis*. Thus, it was found that bengamide A (**1**) was able to block the bacteria growth of this serious pathogenic agent. Therefore, this relevant finding broadens the therapeutic applications of the bengamides as new antibiotics with a novel mechanism of action. These stunning biological discoveries elicited the emerging interest for the bengamides as new potential antibiotics, and as a result, some bengamide analogues have been developed and proved to possess impressive antibiotic activities against various drug-resistant bacteria such as *Mycobacterium tuberculosis* and *Staphylococcus aureus*, expanding the biological applications of these enticing molecules.

The interest of our research group in the design and development of new antitumor agents based on bioactive natural products, led us to consider the bengamides as prime targets, starting a research programme directed to the establishment of an efficient, stereoselective and convergent synthesis that enabled the access to the natural bengamides [30,31,32] and analogs thereof for biological evaluations [33,34,35].

Therefore, due to the recent demonstration of the potential of the bengamides as a new class of antibiotics, we wish to report the advances achieved in this field, including the last progress in the chemistry and biology of these fascinating molecules since 2014, when we reported an extensive review about the bengamides and bengazoles.

The present review reports all these new discoveries, describing in detail the evolution of the bengamides from their initial exploration as antitumor agents up to their current status as antibiotics, highlighting the manifold value of these marine natural products as valid hits in medicinal chemistry.

## 2. Recent Progress in the Chemistry and Biology of the Bengamides

### 2.1. New Progress in Antitumor Properties of the Bengamides

After the publication of our review about the chemistry and biology of the bengamides and bengazoles in 2014 [1], a new review was published in 2017 by Crews et al. [36]. In summary, until 2014, a total of 111 analogues of the bengamides were synthesized [27,37,38,39,40,41], together with numerous total syntheses of the natural congeners [42,43,44,45,46], by different research groups, and biologically evaluated against different tumor cell lines, which, overall, represents an extensive and thorough structure-activity relationship study that has allowed for the establishment of a consistent and well-defined pharmacophore for the bengamides, summarized in the following points: (a) the importance of the substituent at the terminal olefinic position; (b) the essential role of the polyketide fragment, whose hydroxyl groups and stereochemistries are essential for their biological properties; and; (c) the beneficial impact of the modification of the caprolactam fragment in their antitumor properties. Among the analogues that displayed improved pharmacological properties in comparison with the natural counterparts, were the bengamide A analogue **25**, known as LAF389, which presented a greater solubility in water with respect to bengamide A (**1**), or the modified caprolactam analogue **26**, the cyclopentyl analogue of bengamide E **27**, or the ring-opened bengamide analogue **28**, which exhibited a major antitumor potency compared with bengamide E (**5**), being analogue **28** the most potent analogue identified so far (Figure 3).

In 2015, Brönstrup and coworkers isolated bengamides E (**5**), F (**6**), E’ (**7**) and F’ (**8**) from the terrestrial myxobacterium *Myxococcus virescens* ST200611 [13], being the second time that this class of supposedly exclusive marine natural products were isolated from these myxobacteria, after the isolation and characterization in 2012 by Crews et al. of bengamides E (**5**), F (**6**) and E’ (**7**) from a similar specimen of *Myxococcus virescens* [12]. Interestingly, in the relevant contribution of Brönstrup, their authors deciphered the genetic blueprint for the biosynthesis of the bengamides, identifying a polyketide synthase (PKS)/nonribosomal peptide synthetase (NRPS) hybrid system as the responsible for the biosynthesis. Accordingly, the biosynthetic proposal was that an isobutyryl-CoA starter unit was sequentially assembled with a malonyl-CoA, two glycolate units and, finally, a L-lysine residue. After reductive processes of the resulting polyketide, *O*-methylation and a lactamization process, the resulting natural products are produced, which were optionally further *N*-methylated at the caprolactam moiety.

Additionally, the authors disclosed that two copies of methionine aminopeptidases (MetAP1a and MetAP1b) were encoded, whose sequence analyses revealed the presence of a leucine residue in position 154 in contrast to the cysteine or alanine residues present in other prokaryotic and eukaryotic MetAPs, which might explain the resistance that the MetAPs, produced by these myxobacterial, present to the biosynthesized bengamides. On the other hand, the identification of the biosynthetic pathway lets the authors exploit these myxobacterial strains for the production of bengamides and analogues by fermentation. In fact, the bengamide analogues **29**–**33** were prepared by semisynthesis from natural bengamide E’ (**7**) via *N*-alkylation of its per-acetylated derivative (Figure 1a). The biological evaluation of these analogues against a HCT116 tumoral cell line, revealed potent antitumoral activities in the nM range. In pursuit of more potent analogues with potential clinical applications, the authors prepared by total synthesis a series of fused benzocaprolactam derivatives **36**–**41** (Figure 1b), whose biological evaluation against a panel of 14 cancer cell lines furnished excellent results, being the analogues **36** and **39** the most active (Table 2). For the synthesis of these derivatives, the authors employed the synthesis developed by Kinder [27], according to which lactone **34** was reacted with the amino benzocaprolactames **35a**–**f** in the presence of sodium 2-ethylhexanoate to furnish the final bengamide derivatives in 30–55% yields. Analysis of their pharmacokinetic properties showed that these compounds were stable in plasma and had low metabolic labilities when subjected to mouse, rat and human liver microsomes. Furthermore, compound **36** displayed moderate clearance and volume of distribution, a terminal half-life of 3.4 h and a solubility of 370 μM at pH 7.4. An in vivo study of this compound **36** with female C57BL/6 mice bearing early-stage B16 melanoma determined that in the highest nontoxic dose, established at 60 mg/kg per injection, with a total dose of 480 mg/Kg, the antitumor activity was of a 31% for the tumor/control value.

Based on the excellent and striking antitumor properties of some of the described analogues, in our research group, we decided to evaluate the viability of one of their most potent analogues described so far, analogue **28**, against colorectal cancer (CRC) cell lines as a new alternative treatment of colon cancer [35]. A preliminary antitumor evaluation against different tumor and non-tumor cell lines revealed this analogue as a very potent antitumor compound in the low micromolar range, although not so potent as previously described by Nan et al. (Table 3). In addition, it was important to stand out the IC_50_ of **28** against the CCD18 normal colon cell line, which proved to be a 70-fold higher concentration in comparison with the IC_50_ against the T84 and SW480 colon cancer cell lines. This is an indication of a low toxicity of **28** in normal cells compared to tumor cells. In fact, the toxicity of this analogue was evaluated by means of a hemolysis test using human erythrocytes, detecting a very low level of hemolysis at the highest doses of this compound, not causing erythrocyte agglutination, which implies a lack of hemotoxicity. Similarly, an absence of toxicity was confirmed against white blood cells and macrophages, representing a promising option for the treatment of CRC due to its great biocompatibility with blood cells, including cells of the immune system. Currently, we are conducting in vivo studies with this analogue that will allow us to define and determine the potential viability of this compound for its clinical use.

Among the new analogues prepared in the last years, Pham and coworkers have described the preparation of epimers at the caprolactam moiety, which surprisingly proved to be more potent against different tumor cell lines compared with the derivatives with the configuration of the natural products [47]. Thus, after the demonstration that changes of the stereochemistries at the polyketide chain, through the 2-, 2,3- and 3,4-epimers (**42**–**44**) and the enantiomer of bengamide E [48], resulted in almost complete loss of antitumor activities, the 2′-epimer of bengamide E, analogue **45**, exhibited more potent cytotoxicity against six cancer cell lines, namely KB (mouth epidermal carcinoma cells), HepG2 (human liver hepatocellular carcinoma cells), LU (human lung adenocarcinoma cells), MCF7 (human breast cancer cells), HL60 (human promyelocytic leukemia cells) and HeLa (human cervical carcinoma cells), compared with the natural bengamide E (**5**). Similar observations were obtained with the 2′*R* analogues **47** and **48**, including the *N*-substituted caprolactam series **54b**–**56b**, including an interesting selectivity against some cancer cell lines, such compounds **45** or **55b**, which selectively inhibited MCF-7 cells, while **54b** or **56b** displayed a major selectivity toward Lu-1 and HepG-2 cell lines, respectively, with some of them with IC_50_ values less than 1 μM [49].

On the other hand, the corresponding [6]-membered ring derivatives **50**–**53** displayed in general less antitumor activities compared with the [7]-membered caprolactam ring counterparts (Figure 4 and Table 4).

From a synthetic point of view, it is interesting to stand out that all these bengamide analogues were prepared by coupling the lactone **57** with a collection of 19 amines (**58**) with the assistance of microwave irradiation instead of conventional basic conditions, to smoothly afford the resulting amides in reasonable and reproducible yields (Figure 2).

### 2.2. Antifungal Activities of the Bengamides

The biological screening of the bengamides showed that these compounds were inactive against fungi, including *Candida albicans* and *Saccharomyces braziliensis*. Conversely, the bengazoles, which are the second most common secondary metabolites found in *Jaspis* species (Figure 5), stand out for their antifungal activity against *Candida*
*albicans* [50,51], displaying comparable values of inhibition with amphotericin B and with a similar mode of action [52]. In a recent study conducted by Molinski et al. [53], it was found a synergistic activity of a mixture of bengamide A (**1**) and bengazoles A–G (**59a**–**g**). In fact, it was observed that while pure bengazoles presented a MIC value of 1 μM against *C. albicans* at 0.5 μg/disk, inducing a zone of inhibition of 9–10 mm, a sample containing a mixture of bengazoles and bengamides produced a larger inhibition zone of 40 mm. Particularly, in such studies, a mixture of bengazoles A–G, at a constant loading of 0.5 μg/disk, and variable amounts of bengamide A (**1**) were combined and tested against *Candida albicans* ATCC14503. Unexpectedly, in a mass ratio of 400:1 for **1**:**59**, complete inhibition of the antifungal activity of **59** was detected. Then, when the concentrations of bengamide A (**1**) were lowered, a dose-dependent response was observed, finding out that in a 2:1 ratio of **1**:**59** the zone of inhibition increased a 50%, confirming a synergy between both natural products.

### 2.3. Antiviral Activity of the Bengamides

In a recent campaign for the discovery of HIV-1 inhibitors from marine natural products, the Brockman’s group has demonstrated for the first time the potential of bengamide A (**1**) as a new antiviral agent, by inhibition of the NF-kB-dependent HIV-1 replication [54]. Importantly, this study represents the first report of antiviral activity of bengamides, identifying bengamide A as the most potent natural product as a NF-kB inhibitor to modulate HIV-1 replication [55]. Thus, the screening of 252 pure compounds derived from marine invertebrates and microorganisms, through a multicycle HIV-1 replication assay, identified bengamide A (**1**) as the most potent inhibitor, with an activity of EC_50_ around 15 nM, an activity which is comparable to common antiretrovirals such as indinavir, efavirenz and raltegravir. Furthermore, these biological studies revealed that the HIV-1 inhibition by bengamide A (**1**) was not primarily attributable to its cytotoxicity, but this inhibitory capacity was mediated by its effect in the NF-κB-dependent viral gene expression, preventing transcription of HIV downstream of RNA viral reverse transcription and integration. Anyway, as pointed out by the authors, it was not excluded that bengamide A (**1**) could interfere with other steps of the viral replication cycle, which is currently under investigation.

## 3. Development of the Bengamides and Their Analogues as New Antibiotics

As mentioned in the Introduction Section, the preliminary biological evaluations of the bengamides, accomplished by Crews, demonstrated that bengamides A (**1**) and B (**2**) exhibited antibiotic activities against *Streptococcus pyrogenes*. However, no more biological studies were achieved in this direction until 2011 when Ye et al. confirmed the ability of the bengamides to inhibit MetAp of *Mycobacterium tuberculosis* [56]. Taking into account that methionine aminopeptidases are ubiquitous enzymes either in eukaryotic or prokaryotic cells, it was proved that these enzymes are essential for bacteria, leading to their death when MetAP gene is deleted [57]. Importantly, despite that all MetAPs are homologous in sequence in the catalytic domain, mammalian MetAPs present an extension at the *N*-terminus. Although this *N*-terminal extension in human MetAPs is not required for enzyme activity, it could present an important impact on the interaction of the enzyme with inhibitors, and this difference could be exploited for selective inhibition of the MetAP of bacteria over the human MetAPs. Thus, in the particular case of the MetAP of *Mycobacterium tuberculosis*, there are two genes (mapA and mapB in the H_37_Rv genome and map_1 and map_2 in the CDC1551 genome) that express *Mt*MetAP1a and *Mt*MetAP1c, respectively. In the case of *Mt*MetAP1c, this protein was purified, and its structural analysis revealed an SH3 binding motif in its *N*-terminus. On the other hand, the *Mt*MetAP1a is shorter at the *N*-terminus and has no such SH3 binding motif. In both MetAPs, the active site of the enzyme is a pocket with a hydrophobic environment where is situated a divalent metallic ion such as Mn, Fe, Co or Ni. Next to the pocket hole is situated a mobile hydrophobic loop whose interaction with the substrate is believed to be weak, but not clear if there is an actual interaction [58].

In this way, most residues in the active site pockets of MetAPs of both prokaryotic and eukaryotic origin are conserved. However, at the opening and outside of the pocket, more structural variations exist because of the various lengths and residues in the *N*-terminal extension, opening the possibility of exploring additional interactions that could potentially differentiate the inhibition potency between the bacteria and mammalian peptidases [59]. According to the molecular framework of the bengamides and, taking into account their mode of interaction at the active site of the MetAPs, structural modifications that could modulate the selectivity in the inhibition should be done in the amide fragment, replacing the caprolactam moiety by various amide moieties that could generate additional interactions to display inhibition against tubercular MetAPs and weak or no inhibition of human MetAPs. Aiming to find high inhibition potency and selectivity against different cell MetAPs, the Ye group achieved the synthesis of a set of seven analogues **60**–**66**, which were evaluated as potential antitubercular agents (Figure 6) [56].

In these biological evaluations, their inhibitions were measured against purified *Mt*MetAP1a and *Mt*MetAP1c, which were activated by divalent metal ions including Co^II^, Mn^II^, Ni^II^ and Fe^II^ [60]. For therapeutic applications, it is crucial that the inhibition assays are done in its native metalloform because inhibitors can display marked variations in inhibitory potency toward different metalloforms of the enzyme, as has been demonstrated in studies with *E. coli* MetAP [61,62,63]. According to the obtained results, as summarized in Table 5, compounds bearing aromatic rings (**60**, **61** and **66**) showed remarkable potencies toward the Co^II^, Mn^II^ and Fe^II^ forms of *Mt*MetAP1a, probably due to additional interactions of the aromatic systems.

The obtention of crystals of the complexes of *Mt*MetAP1a with either **65** or **66** let the obtention of the X-ray structures, confirming that both inhibitors interact in the same way as bengamides with human type 2 MetAP, according to which the triol system coordinates with the two active site metal ions, and the *tert*-butyl alkene chain occupies the P1 pocket through a hydrophobic interaction [64]. In the case of **65**, in which the amide moiety is shorter, this fragment takes a position similar to that of the caprolactam ring of the bengamides. Interestingly, the trimethylphenyl group of **66** is embedded in the shallow cavity at the opening present, particularly in the MetAP of the *M. tuberculosis*. In addition, these inhibitory activities were associated with an antitubercular activity, being analog **66** the most active against both replicating *M. tuberculosis* with a MIC value of 50.6 μM (comparatively, rifampin has a MIC value of 0.08 μM and isoniazid with a MIC value of 0.24 μM) and non-replicating *M. tuberculosis* with a MIC value of 107.4 μM (comparatively, rifampin has a MIC value of 1.96 μM and isoniazid with a MIC value of >128 μM) (In these studies MIC values were defined as the percent inhibitions at 128 μM or as the lowest concentration, effecting a decrease of ≥0% in fluorescence or luminescence assays relative to untreated controls).

For the development of some of these analogues as new antitubercular agents, it is essential to minimize inhibition of human MetAPs to decrease of this way the toxicity. To assess the potential toxicity of these analogues, their effects on the growth of human K562 cells were evaluated with the result that analogues **62**, **63**, **64** and **65** displayed no or weak activity, being **66** the most active against the growth of these human cancer cells. These results demonstrated that the selectivity of these bengamide-based antibiotics required to be significantly improved.

Following these pioneering studies of Yu et al., as part of a research programme directed towards the identification of new leads against *M. tuberculosis*, 1434 diverse marine extracts were screened for their ability to inhibit *M. tuberculosis* growth in vitro [65]. From this extensive screening, 18 extracts, 11 from the Porifera phylum and 5 from the Chordata phylum, were identified with the ability to inhibit the growth of *M. tuberculosis* in a 50-0.39 μg/mL range for MIC_50_, defined as the concentration that resulted in 50% survival of bacteria in comparison to untreated controls. Among all these extracts, two samples from Porifera derived from a *Tedania* sp. sea sponge from Queensland in Australia, displayed outstanding MIC_50_ values of 0.39 and 1.56 μg/mL, respectively. The purification of the active ingredients of both extracts led to the identification of bengamide B (**2**) as the responsible for this antitubercular activity. Having bengamide B (**2**) in hand, the authors evaluated its inhibitory capacity against purified MetAP enzymes derived from *M. tuberculosis* (*Mt*MeAP1c), *E. faecalis* (*Ef*MeAP1b) and human MetAP (*Hs*MeAP1b), proving to be a highly potent inhibitor for the three enzymes with inhibition percentages of 72.84, 83.33 and 84.72%, respectively at a single concentration of 50 μM of bengamide B (**2**). For the particular case of *Ef*MeAP1b, it was possible to check that, despite its strong inhibition of this enzyme, this activity was not reflected in bacterial inhibition in vitro studies against *E. faecalis*.

On the other hand, despite its inhibition against human MetAP (*Hs*MeAP1b), bengamide B (**2**) did not result in cytotoxic against different human cells, including THP-1, HepG2, HEK293 and A549. More interesting and fruitful was the study of combination therapy against tuberculosis (TB) by use of bengamide B (**2**). Accordingly, a combined therapy consisting of bengamide B (**2**) and rifampicin against *M. tuberculosis* H_37_Rv revealed a strong synergy according to the Chou–Talalay combination index (CI), which was 0.1–0.3, allowing for a dose reduction index (DRI) from 8-fold to over 200-fold for rifampicin and 3-fold to over 14-fold for bengamide B (**2**).

More recently, Nan et al., who designed, synthesized and evaluated a wide library of ring-opened bengamide analogues as antitumor agents, paid attention to the development of new antibacterial compounds based on bengamides [66]. To this aim, they synthesized a first library of the new ring-opened bengamide analogues **68a**–**f** by coupling the key precursors **57** and **67****a**–**f**, according to the synthetic route depicted in Figure 3. This set of compounds was then tested against *Staphylococcus aureus*, strain 8325-4, resulting active enough according to the MIC values obtained, as indicated in Figure 3. This biological evaluation was then extended to the following *S. aureus* strains, including *S. aureus* clpP mutant strain (ΔclpP), *S. aureus* clpP complementary strain (ΔclpP::clpP) and the 8325-4 strain overexpressing clpP (OEclpP). These bacteria strains are characterized by overexpressing the protease caseinolytic protease P (ClpP) [67], which is an energy-dependent serine protease, which plays an essential role in bacterial pathogenesis [68].

These proteases are actually present in either bacteria or in eukaryotes and allow the cell to maintain proteostasis by controlling the degradation of undesired proteins from the intracellular environment, including those involved in regulating stress responses and virulence factor production. In the case of *S. aureus*, it was well established that *S. aureus* ClpP (SaClpP) regulates bacterial virulence and pathogenesis, playing a critical role in infectivity and virulence during host infection. Not surprisingly, ClpP has attracted the interest of the medicinal chemistry community as a novel valid target for antibiotic discovery. As a consequence, small molecules have been described capable of dysregulating the function of ClpP, towards either its inhibition or activation, which leads to a bactericidal activity [69]. In the particular case of ClpP activators, their effect is the bacteria inhibition or death [70], whereas ClpP inhibitors produce a decrease in bacterial virulence [71]. Thus, the first ClpP activators corresponded to the named acyldepsipeptides ADEPs, such as ADEP-1 or ADEP-4, whose discovery was the result of a high-throughput screening study [72,73]. As a consequence of the ClpP activation, the process of protein degradation is out of control, affecting unfolded not only proteins but also nascent proteins, which triggers the inhibition of cell division and eventually cell death. Interestingly, when the bengamide analogues were evaluated against the aforementioned ClpP-related strains of *S. aureus*, some of them, such as **68a**, showed potent activity against OEclpP strain with a MIC value of 0.78–1.56 μg/mL, which could be an indication that bengamides might activate direct or indirectly ClpP.

In order to increase the antibiotic activity of this class of bengamide analogues, further structural optimization was accomplished by modifying the size of the terminal rings and by replacement of the ester group with a linear alkynyl motif to improve biological stability due to the known instability and sensitivity of the esters to the metabolism [74]. This structural optimization led to compound type **69**, and then the second generation of ring-opened bengamide analogues was prepared by modification of the methyl group corresponding to the alanine residue by different alkyl and aryl groups (**69a**–**f**) (Figure 7).

The biological evaluation against the different strains of *S. aureus* of this second series revealed that compounds **69a** and **69f**, with MIC values of 1.56–3.13 and 6.25–12.5 μg/mL, respectively, were able to activate ClpP to degrade α-casein [75], being compound **69f** more active for *Sa*ClpP activation. Then, the third round of optimization was achieved by changing the *para*-fluorine group contained in compound **69a** to various substituents at different sites on the phenyl ring, preparing the series of compounds **70a**–**l**. From this new library of bengamide analogues, compound **70i** (MIC = 6.25–12.5 μg/mL) displayed moderate activity for *Sa*ClpP activation in vitro. Finally, the *Sa*ClpP activation and antibacterial activity observed in compounds **69f** and **70i** allowed the authors to design and synthesize compound **71**, which displayed activation on the degradation of protein substrate by *Sa*ClpP with an EC_50_ of 3.13–6.25 μM, while was still active against *S. aureus* in a ClpP-dependent manner (Figure 7). Therefore, the authors proved that bengamide analogues might potentially act as a new class of antibiotics by targeting the *S. aureus* ClpP protease.

As a continuation of the research previously described, one year later, Nan and co-workers reported the synthesis of a new series of ring-opened bengamide analogues [76] in order to improve the antibacterial activity displayed by the analogue **68a** and to explore the antibiotic potential against methicillin-resistant *Staphylococcus aureus* (MRSA), which represents a real threat for the human health as current antibiotics are no longer effective against these bacteria [77]. Thus, taking **68a** as the hit compound, they synthesized a total of twenty-four bengamide analogues with opened rings (**72a**–**g**, **73a**–**m** and **74**–**76**) (Figure 8) in two phases. In the first phase, they prepared the series **72a**–**g** to assess the contribution of the alanine residue to the anti-MRSA activity by replacement of the methyl group with different alkyl groups and by the change of the stereochemistry at this position. To this aim, these compounds were tested against six different kinds of MRSA, namely USA300, NRS-1, NRS-70, NRS-271, NRS-108 and NRS-100, and using vancomycin (**Van**) and tetracycline (**Tetra**) as reference drugs (Table 6).

The biological results of this study reflected that the replacement of the methyl group of *L*-alanine did not improve in any case the anti-MRSA activity. In light of these results, the authors decided to keep the *L*-alanine residue and, then explore the effect of the ester group on the antibacterial activity. In this second phase, the first set of 7 different esters **73a**–**g**, were prepared, and it was proved, after the corresponding biological evaluations, that bulky aliphatic groups presented a beneficial impact in their anti-MRSA activities. These interesting results encouraged the authors to prepare the adamantyl derivatives **73h**–**m** as a suitable and rational choice. From this study, the authors identified **73j** as the most potent analogue, with MIC values against the different MRSA strains in the 0.04–0.31 μg/mL range.

Given the poor stability of the ester functional group in organisms, the corresponding amide **74** was synthesized and biologically evaluated. However, to the dismay of the authors, this compound was much less potent than its ester analogue, with a MIC value of 2.0–4.0 μg/mL, which indicated that the ester group seemed to be crucial for the anti-MRSA activity. Similarly, the terminal olefin analogue **75** and the reduction product **76** were prepared, but their biological activities against MRSA remarkably decreased, being their MIC values of >50 and 1.00–8.00 μg/mL, respectively (Table 6). Therefore, given the excellent antibacterial activity of the adamantane ethanol ester derivative **73j** towards the six *Staphylococcus aureus* strains previously cited, its pharmacokinetic parameters were evaluated by means of an in vivo assay, according to which **73j** was provided to mice via oral or intraperitoneal administrations in doses of 100 mg/Kg. The analysis of peripheral blood at different times showed that **73j** was hydrolysed to its corresponding acid from the beginning. This poor stability in mouse plasma was ascribed to the high levels of carboxylesterases. In contrast, **73j** displayed great stability in human plasma and kept its anti-MRSA activity with a MIC of 0.5 μg/mL on the USA300 strain. In addition, compound **73j** did not exhibit inhibitory activity against hERG (IC_50_ > 40 μmol/L), which is a promising indication of clinical safety.

Finally, as mentioned in the Introduction section, very recently, a new member of the bengamides, bengamide R (**19**), was isolated and characterized from *Jaspis splendens* sponges, collected in Mauritius, as part of a research programme directed towards the discovery and identification of new antimycobacterial agents from marine sources [16]. Due to the severity of tuberculosis (TB), caused by multidrug-resistant (MDR) and extensively drug-resistant (XDR) Mtb strains, the development of effective therapies is urgently needed, and as a consequence, the search and identification of new anti-TB agents are required [78]. Given the impressive and extensive source of bioactive metabolites that represents marine organisms [79], the authors decided to tackle an extensive screening, in which a wide variety of 984 marine invertebrate extracts were evaluated, detecting two extracts against *Mycobacterium tuberculosis* corresponding to the sponges *Hyrtios reticulatus* and *Jaspis splendens* as the most active with MIC_90_ (Minimum concentration in μg/mL to inhibit the growth of the bacteria a 90%) values of 51 and 2.4 μg/mL, respectively. While for the *Hyrtios* extract, its activity was associated with the natural product heteronemin [80,81], for the *Jaspis* extract, it was demonstrated that the responsibility for the antimycobacterial activity was the bengamides P (**17**) and Q (**18**), which were isolated and purified from this extract. These results provide more evidence of the potential of the bengamides as promising new antibiotics and, in particular, as novel antituberculosis leads. 

## 4. Conclusions

The discovery, structural elucidation and biological properties of the bengamides, a family of natural products isolated from *Jaspis* sponges and also from the terrestrial myxobacteria *Myxococus virescens*, elicited intense research activity, attracting the interest of chemists and biologists alike during the last decades due to their potent antitumor properties with a fumagillin-like mechanism of action by inhibition of the methionine aminopeptidases. As amply demonstrated in the literature, bengamides are a very interesting family of natural products with an outstanding biological profile, which includes not only potent antitumoral activity but also antihelmintic, antiviral and antibiotic activities. The manifold value of the bengamides, together with their unique mechanism of action and unprecedented molecular architectures, has positioned them as excellent targets in medicinal chemistry and have provided great opportunities in this field. As a consequence, enormous efforts from different research groups around the world have been made directed towards their total synthesis, design of new analogues and biological evaluations, involving around more than 150 synthetic bengamide analogues up to date. 

Overall, this investigation led to important new insights and useful structure-activity relationships within the bengamide family that allowed for the identification of new chemical entities with powerful potencies and improved pharmacological properties beyond those of the naturally occurring bengamides. Interestingly, in recent years, the potential of the bengamides as antibiotics has been explored and investigated, being witnessing an evolution of the bengamides from their initial consideration as antitumor agents up to their current status as promising antibiotics. In this review, we had reported the recent progress in this new direction of the bengamides, together with the new advances in the antitumoral, antifungal and antiviral properties of these enticing molecules since 2014, when we reported an extensive review of these natural compounds.

In this sense, the growing emergence of drug-resistant bacteria strains and the lack of new antibiotics is currently one of the most serious problems in human health. Thus, the search for new antibiotics must become a maximum priority for the scientific community. The new discovery of the bengamides as potential antibiotics opens promising opportunities in the future that might deal with this urgent problem. However, although massive investigations have been devoted to the begamides, still persistent research has to be carried out to position bengamide analogue candidates in the clinic, and this must be the priority of medicinal chemistry research groups in the incoming years.

## Data Availability

Not applicable.

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
