# Peer review of "The Development of the Bengamides as New Antibiotics against Drug-Resistant Bacteria"

_marinedrugs, 2022, doi:10.3390/md20060373_

Round 1

Reviewer 1 Report

Porras-Alcalá et al. have submitted an interesting and timely review on bengamides as potential new antibiotics. It should be suitable for publication in Mar Drugs pending some revisions.

  1. Page 4, lines 98-101 state that LAF-389 was discontinued in a phase I clinical trial due to toxicity effects. Please provide more details on these toxicities, particularly given that a fatality was recorded during the study. The results of this study should also be considered with respect to the potential and likelihood of future clinical studies for antibiotics and other indications using this chemical class.
  2. In lines 158-161 of page 6, it’s stated that the most potent analogues with improved properties over the natural products include compounds 25-28, as shown in Figure 3. However Figure 3 shows that 25-28 have weaker activities than compounds 1, 2, 5, 6. Please clarify this discrepancy.
  3. Page 10, lines 282-283: the effects of bengamide A on NF-κB-dependent viral gene expression do not “produce the downstream of HIV reverse transcription.” Rather this prevents transcription of HIV that has already integrated into cells. Please fix this.
  4. Please define MIC at its first occasion, which I think is page 12, line 341. Also on Figure 7, MIC is defined as Minimum inhibitory concentration or percent inhibition at 128 uM (IC50). This is confusing as these values could reflect MIC, percent inhibition at 128uM, or an IC50. Please clarify.
  5. Page 13, line 362 says that Porifera extracts displayed values of 0.39 and 1.56 ug/mL, what is being measured here?
  6. Lines 367-368 say that compound 2 was a highly potent inhibitor for the three enzymes with values of 72.84, 83.33 and 84.72% - at what concentration or by what measure?
  7. From lines 421-430, please provide in the text the values for activities observed for each compound. I see the activities are listed in Figure 8 but the paragraph will be much clearer if activities are also stated here for the selected compounds.
  8. Similarly, for reader clarity, please provide in the text the exact values for activities for compounds described on page 15, lines 448-466.

Reviewer 2 Report

The review is really interesting from a biological focus, but it is lacking a section where the synthetic chemistry should be outlined.   I would like to suggest to add a section where the main synthetic strategies are discussed.  This will be a great improving to this already excellent review.

The authors show a little “ego” outlining what its group has been done.  Probably, these parts written in first person can be removed or at least modified.  

The paragraph #100-116 is a little confusing

# 105  …. However, their antibiotic properties were not further explored.

# 110 ….. and then, broadening the therapeutic applications of the bengamides as new antibiotics with a novel mechanism of action.

Maybe breaking the paragraph in two and adding an adverb at the beginning of the second paragraph will clarify it

The paragraph #116-136 could be shortened.  It contains historical information of one of the research groups, which is not of interest to the general reader

#137-140.  Is only the work of the group is discussed? Or the work of other groups are also discussed? 

#143-147 The difference between the two reviews is not interesting at all

Definition of bengamide analogues should be discussed.  Could the compounds that have substituted the caprolactam by an aromatic ring considered bengamide analogues?

Reviewer 3 Report

  1. Why authors said bacteria in the title, while the study describes much about anticancer, antifungal, and antiviral properties?
  2. The abstract should be more focused on the details in the manuscript.
  3. I found this review biased more towards structural diversity and synthesis of bengamides, however, it is important equally to describe the mechanisms of actions for the development. Please update the manuscript accordingly.
  4. Authors need to discuss more about the cytotoxicity of bengamides in different models such as cell lines or In vivo models. Bengamides reported for a long, and there should be many studies about all this. Please justify and update.
  5. What about apoptosis? Usually, anticancer agents are apoptotic, please detailed about bengamides.
  6. What about the clinical status of bengamides? Have they been ever in a clinical trial? Please include a table showing different bengamides in clinical studies, and studies showing in vivo efficacy of bengamides. Please update.
  7. Authors need to include figures representing, the biological activities and mode of actions of bengamides. Please update.
  8. Please detail the legend. Which organism’s methionine aminopeptidase, are the authors talking about?
  9. Do benamides show cytotoxicity against normal cell lines? Please check and update.

Round 2

Reviewer 3 Report

The authors failed to respond to the reviewer's comments or suggestions. 

Author Response

We appreciate the reviewer´s  comments, which have been very useful to further improve quality of the original article. According to the response, it is possible that our response to his/her comments, included as a attached document, was not received. Please, we provide a point-by-point response to the comments in the enclosed document.
